# Cell-Main Spectra Profile Screening Technique in Simulation of Circulating Tumour Cells Using MALDI-TOF Mass Spectrometry

**DOI:** 10.3390/cancers13153775

**Published:** 2021-07-27

**Authors:** Wararat Chiangjong, Sebastian Chakrit Bhakdi, Noppawan Woramongkolchai, Thitinee Vanichapol, Nutkridta Pongsakul, Suradej Hongeng, Somchai Chutipongtanate

**Affiliations:** 1Pediatric Translational Research Unit, Department of Pediatrics, Faculty of Medicine Ramathibodi Hospital, Mahidol University, Bangkok 10400, Thailand; wararat.chi@mahidol.ac.th (W.C.); nutkridta.pon@mahidol.ac.th (N.P.); 2Department of Pathobiology, Faculty of Science, Mahidol University, Bangkok 10400, Thailand; sebastianchakrit.pun@mahidol.ac.th (S.C.B.); 6271011133@student.chula.ac.th (N.W.); 3X-ZELL Biotech Pte Ltd., Singapore 069535, Singapore; 4Department of Pharmacognosy and Pharmaceutical Botany, Faculty of Pharmaceutical Sciences, Chulalongkorn University, Bangkok 10330, Thailand; 5Division of Hematology and Oncology, Department of Pediatrics, Faculty of Medicine Ramathibodi Hospital, Mahidol University, Bangkok 10400, Thailand; tvan147@aucklanduni.ac.nz (T.V.); suradej.hon@mahidol.ac.th (S.H.); 6Department of Clinical Epidemiology and Biostatistics, Faculty of Medicine Ramathibodi Hospital, Mahidol University, Bangkok 10400, Thailand; 7Chakri Naruebodindra Medical Institute, Faculty of Medicine Ramathibodi Hospital, Mahidol University, Samut Prakan 10540, Thailand

**Keywords:** cell main spectra, circulating tumour cell, MALDI-TOF, method development

## Abstract

**Simple Summary:**

Cancer cells can detach from a primary tumour and present in peripheral blood as circulating tumour cells, or in the widest sense, as circulating atypical cells (CAC). Although CAC are a promising biomarker for non-invasive cancer screening, they occur at very low frequency and their detection and characterization remains challenging. We here validated isolation and concentration of untouched CAC from spiked cancer cell blood samples and combined this with matrix-assisted laser desorption ionization–time of flight mass spectrometry (MALDI-TOF MS). This workflow was optimised to detect as little as six cancer cells per 5000 white blood cells. Future development of our workflow may cover a larger range of cancer types and further improvements to enable the use of MALDI-TOF MS as a cancer-screening platform in clinical settings.

**Abstract:**

Circulating atypical cells (CAC) are released from a primary tumour site into peripheral blood and are indicators of cancer metastasis. CAC occur at very low frequency in circulating blood, and their detection remains challenging. Moreover, white blood cells (WBC) are the major contaminant in enriched CAC samples. Here, we developed matrix-assisted laser desorption ionization–time of flight mass spectrometry (MALDI-TOF MS) as a novel CAC characterization platform. Main spectra profiles (MSP) of normal and cancer cells were generated by MALDI-TOF MS, and a cell-main spectra database was then compiled and analysed using the MALDI Biotyper software. Logarithmic scores accurately predicted distinct cell types. The feasibility of this workflow was then validated using simulated samples, which were prepared by 5000 WBC of three healthy individuals spiked with varying numbers (3, 6, 12, 25, 50, and 100) of lung, colon, or prostate cancer cells. MALDI-TOF MS was able to detect cancer cells down to six cells over the background noise of 5000 WBC with significantly higher predictive scores as compared to WBC alone. Further development of cell-MSP database to cover all cancer types sourced from cell lines and patient tumours may enable the use of MALDI-TOF MS as a cancer-screening platform in clinical settings in the future.

## 1. Introduction

Early detection of aggressively growing, clinically significant cancers remains challenging. Tissue invasion and metastasis is one of the hallmarks of cancer, and today, it is widely assumed that metastasis happens early in aggressively growing tumours. Therefore, it is fair to assume that detection of tumour cells dislodged from the primary tumour may be able to contribute to early diagnosis. Various subtypes of cells from the primary tumour have been described to shed or migrate into the circulation. Cells from the actual tumour tissue may undergo epithelial–mesenchymal transition followed by transendothelial migration, resulting in haematogenic metastasis of epithelial-derived circulating tumour cells. Another cell type observed in the blood of cancer patients are atypical, tumour-associated circulating endothelial cells (tCEC), shed from the growing blood vessels of the tumour—in other words, from sustained angiogenesis, another important hallmark of cancer [1,2,3,4,5]. Independent of their particular subtype, cancer cells enter and survive in the blood circulation, as single cells or in aggregates, for a long time before the diseases are diagnosed [6,7]. In an attempt to account for the large diversity of cells shed into the blood of cancer patients, the present study will refer to all tumour-associated circulating cells as “circulating atypical cells” (CAC).

A substantial number of CAC can shed from a small tumour, but they still present a very low number compared to the number of blood cells, estimated at around one CAC per 7.5 mL whole blood [8]. The complexity of peripheral blood also hinders CAC detection. The use of diagnostic leukapheresis, which allows sampling of larger blood volumes (up to 2.5 L) and concentrates CAC together with mononuclear cells, has been proposed instead of using whole blood [9,10,11]. Nonetheless, detecting CAC in the early stages of tumour development holds promise as a non-invasive liquid biopsy for cancer screening. Many attempts have been made to overcome these limitations by improving technologies for CAC enrichment and detection.

Circulating atypical cells can be enriched by (1) making use of the physical properties of CAC, such as size, density or rigidity; (2) depleting contaminating red blood cells (RBC) and white blood cells (WBC) (negative isolation); or (3) immunological targeting of unique surface proteins (positive isolation).

To enrich CAC based on their physical properties, larger and more structurally rigid CAC were isolated by filter membranes with specific pore sizes [12,13]. Density differential centrifugation was also used to separate CAC and WBC from RBC [14].

For CAC enrichment by negative isolation, WBC obtained from whole blood samples were labelled via the CD45 pan-leukocyte antigen and depleted by magnetic separation, and the resulting CD45 negative cell population was analysed for the presence of CAC [1,15].

For positive isolation, epithelial CAC can be labelled via their epithelial cell adhesion molecule (EpCAM) and isolated by magnetic beads [8]. Positive isolation of (non-epithelial) CAC has rarely been attempted, with the exception of labelling of membrane-vimentin [16].

Characterization of CAC in isolates is usually performed by immunostaining of specific markers, such as epithelial, mesenchymal or/and endothelial antigens, in combination with the CD45 pan-leukocyte antigen. Epithelial CAC are characterised using specific fluorophore-labelled antibodies, for example EpCAM and a combination of cytokeratins. At the same time, they demonstrate a lack of the CD45 pan-leukocyte marker [17,18,19]. However, the presence of epithelial markers on contaminating non-malignant cells, a loss of marker expression on CAC, and leukocyte-CAC cell fusion were reported to hamper sensitivity and specificity of these approaches [17,20]. Thorough characterization of mesenchymal or endothelial CAC is particularly challenging as it requires multiplexed staining of a much wider range of cellular antigens [21]. In addition, remaining WBC contaminate the enriched CAC sample to various degrees and tend to create significant background noise that complicates CAC detection in the microscope [22,23,24].

Several other methods can be used to characterise CAC. Real-time PCR-based methods detect tumour-specific transcripts, e.g., *HER2*, *ER*, *MUC*, *KRT19* [17]. However, the high sensitivity of these methods frequently complicates the analysis due to contamination from the legitimate transcription of gene markers or pseudogenes in non-malignant cells [17,25]. In vitro culture of CAC and high-throughput next-generation sequencing can be powerful and precise but, with the current state of technology, these approaches are uneconomical and too time-consuming to be used for routine cancer screening.

Recent advances in mass spectrometry technology have revolutionised biomedical research and clinical laboratory investigation. Matrix-assisted laser desorption ionization–time of flight mass spectrometry (MALDI-TOF MS) has been introduced as a rapid, highly sensitive, readily automated, and cost-effective method for bacterial typing in clinical specimens [26,27,28]. Reliable results depend on the logarithmic predictive scores generated by matching the main spectra profiles (MSP) of the analytical sample to the reference bacterial MSP database using the MALDI Biotyper program. MALDI-TOF MS has been used to discriminate immune cells, such as T-lymphocytes, monocytes, polymorphonuclear cells, monocyte-derived macrophages, and dendritic cells. Therefore, we hypothesised that MALDI-TOF MS can be applied for detection of CAC as well.

In the present study, we generate an MSP database covering blood components and normal blood cells, as well as a variety of primary cells and immortalized cancer cell lines. Further, we validate RBC and WBC depletion of the hMX rare-cell negative isolation platform in the context of MALDI-TOF MS. Last, we determine the limit of detection of cancer cells spiked into WBC obtained from the flow-through of the hMX system, thereby delivering a first proof-of-concept for combining rare-cell negative isolation with MALDI-TOF MS for early detection of clinically significant cancers.

## 2. Materials and Methods

### 2.1. Culture of Immortalised Cell Lines

Human cell lines from American Type Culture Collection (ATCC) including fetal intestinal cells (FHs74Int; accession number: CCL-241), T-cell leukaemia (Jurkat; TIB-152), neuroblastoma (SK-N-SH; accession number: HTB-11, and SH-SY5Y; accession number: CRL-2266), lung adenocarcinoma (A549; accession number: CCL-185), colorectal cancer (Caco-2; accession number: HTB-37), and prostate cancer (LNCaP; accession number: CRL-1740), and ovarian cancer from European Collection of Authenticated Cell Culture (ECACC) (A2780; accession number: 93112519) were maintained in the culture media, as shown in Appendix A, under a humidified atmosphere with 5% CO_2_ at 37 °C for 24–48 h. Adherent cells were washed with phosphate-buffered saline (PBS) once to remove all fetal bovine serum (FBS) before cell detachment by adding 0.25% (*w/v*) trypsin-EDTA solution (Biochrom, Berlin, Germany) and incubating at 37 °C for 5–15 min. After stopping trypsinization by adding complete growth medium, the suspension cells were transferred into a 15 mL tube and centrifuged at 200× *g* for 5 min to collect cell pellets. The cell pellets were washed with PBS twice and resuspended in PBS, and cells were counted in a haemocytometer.

### 2.2. Generation of Activated T-Cells and OKT3/CD28 Blasts

Human peripheral blood mononuclear cells (PBMC) were prepared from 10 mL EDTA normal peripheral blood collection. The protocol was approved by the Human Research Ethics Committee, Faculty of Medicine Ramathibodi Hospital, Mahidol University, based on the Declaration of Helsinki (Protocol ID-06-61-09). The whole blood was diluted in PBS at a 1:1 ratio, and the diluted blood was carefully layered onto the Ficoll-Paque solution (Robbins Scientific Cooperation, Norway) at a 2:1 ratio (diluted blood:Ficoll-Paque solution). The tube containing layer solution was centrifuged at 400× *g*, 20 °C for 35 min with no break. Cells in the PBMC layer were pipetted into a new centrifuge tube.

PBMC were washed with PBS twice before culturing in RPMI-1640, supplemented with 100 U/mL IL-2, 10% FBS and 1% penicillin/streptomycin, for 2 days at 37 °C under a humidified atmosphere with 5% CO_2_ incubator. After collecting PBMC, the suspended PBMC were washed twice with PBS and stimulated to become either phytohemagglutinin (PHA) activated T-cells or OKT3/CD28 T-cell blasts.

PHA-activated T-cells were derived from 1 × 10^6^ PBMC that were cultured in RPMI-1640 containing 5 µg/mL PHA, 10% FBS, and 1% penicillin/streptomycin for 3 days at 37 °C under a humidified atmosphere with 5% CO_2_ incubator. The half volume of the cultured medium was discarded and then replaced with fresh RPMI-1640 cultured medium containing 100 U/mL IL-2, 10% FBS, and 1% penicillin/streptomycin. The cells were further cultured for 3 days at 37 °C under a humidified atmosphere with 5% CO_2_ incubator.

For a generation of OKT3/CD28 T-cell blasts, 1 × 10^6^ PBMC were cultured in an anti-CD3- and anti-CD28-coated plate. In brief, 1 µg each of anti-CD3 and anti-CD28 antibodies were added into 1 mL PBS in a 24-well plate and incubated at room temperature for 2 h. The coated well was washed with PBS once before adding PBMC. The PBMC in the coated well were cultured in RPMI-1640, supplemented with 100 U/mL IL-2, 10% FBS, and 1% penicillin/streptomycin for 3 days at 37 °C under a humidified atmosphere with 5% CO_2_ incubator to obtain OKT3/CD28 T-cell blasts. All culture medium for PBMC, PHA-activated T-cells, and the OKT3/CD28 T-cell blasts were replaced every other day with fresh media containing 100 U/mL IL-2. The suspension cells were transferred into a 15 mL tube and centrifuged at 200× *g* for 5 min to collect cell pellets. The cell pellets were washed with PBS twice and resuspended in PBS, and the cells were counted in a haemocytometer.

### 2.3. Plasma and Red Blood Cell Preparation

Plasma from a healthy blood donor was diluted with distilled water to make two-fold serial dilutions for up to 10 dilutions. Protein concentration for each dilution was measured by Bradford’s assay.

Packed red cells in the bottom layer from the Ficoll-Paque method were collected and washed three times with PBS; 0.5 µL of the RBC pellet was diluted in 1 mL of PBS and then counted using a haemocytometer. RBC were diluted in PBS and dotted 1 μL of RBC suspension (100,000, 50,000, 25,000, 12,500, 6250, 3150, or 1562 cells/μL) on the ITO coated slide.

### 2.4. hMX RBC Lysis and WBC Depletion

Blood was obtained from healthy donors from the cubital vein. RBC lysis and CD45-based high-flow magnetic WBC depletion were performed following the manufacturer’s protocol.

In brief, four volumes of hMX lysis buffer (X-Zell, Singapore) were added to one volume of whole blood. Samples were incubated for 7–10 min (until clear) and washed with PBS containing 5 mM EDTA/1% FBS. Centrifugation was performed at 400× *g* for 10 min at the lowest centrifuge acceleration and deceleration at room temperature. After this, cells were resuspended in 50 µL PBS/EDTA 5 mM/FBS 1% and counted in a haemocytometer.

Ten to fifteen million cells were centrifuged at 10,000 rpm for 1 min, the supernatant was discarded, and 1% FBS in PBS-EDTA was added to make a final volume of less than 50 µL. WBC were then isolated by CD45-based immunoaffinity-magnetic separation as previously described [20]. Cells were blocked with Fc-receptor blocking reagent (BioLegend, San Diego, CA, USA) for 15 min at 4 °C, after which one microlitre each of anti-human CD45 IgG conjugated biotin (EXBIO, Vestec, Czech Republic) and anti-human CD235a conjugated biotin (eBioscience, San Diego, CA, USA) were added into the cell suspension and incubated at 4 °C for 15 min. Cells were then washed once in 1% FBS in PBS-EDTA at 400× *g* for 10 min. One hundred microlitres of hMX anti-biotin nanobeads (X-Zell) were added into the sample and then incubated at 4 °C for 15 min. Then, 1 mL of hMX separation buffer (X-Zell) was added, and the mixture was loaded into an hMX separation column (X-Zell) mounted in an hMX separator (X-Zell). After washing by 20 mL hMX separation buffer, 10 mL of hMX buffer (X-Zell) was loaded into the column, and the flow-through was collected. The flow-through was diluted in PBS with a ratio of 1:1 before centrifugation at 10,000 rpm for 1 min. WBC pellet was washed with PBS twice and resuspended in PBS before cells were counted in a haemocytometer. WBC obtained from the flow-through were then used for spiking experiments with cancer cells.

### 2.5. Recovery Rates of Cancer Cells during hMX Lysis and Depletion

Cancer cells were treated with hMX lysis buffer exactly as described for RBC above. To examine recovery rates after hMX lysis, 9000 to 20,000 cancer cells were resuspended in 4 mL hMX lysis buffer, incubated, washed and resuspended in PBS/EDTA 5 mM/FBS 1%. For recovery rates after hMX depletion, 5000 to 10,000 cancer cells were Fc-blocked, incubated with biotin-anti-CD45 and biotin-anti-CD235a antibodies, washed, incubated with anti-biotin beads, washed again and processed through magnetic separation columns as aforementioned. For higher cell counts, aliquots of cell suspensions were counted in a haemocytometer after each respective procedure. For 100 and 10 cell counts, the total volume of cell suspension after lysis and hMX depletion, respectively, was adjusted to 150 µL, pipetted into the well of a 96-well microtitre plate and cells were counted on an inverted fluorescence microscope at 20× magnification (Olympus, Tokyo, Japan).

### 2.6. Spiked Cancer Cell/WBC Sample Preparation

WBC from an individual obtained from hMX flow-through were counted and diluted in 70% methanol with a final concentration of 5000 cells/μL. Cancer cells including A549, Caco-2, and LNCaP, were counted and diluted in 70% methanol in serial dilution with final concentrations of 100, 50, 25, 12, 6, and 3 cells/μL. These serially diluted cancer cells were spiked into WBCs (5000 cells/μL) before dotting on the ITO coated slide. Serial dilutions were performed from stock solution. Aliquots from each serial dilution were counted in triplicate. A forth aliquot of the respective serial dilution was used in the experiment.

### 2.7. Cell Dotting on ITO Coated Slide

Each cell type in PBS was counted using a haemocytometer. One million cells were aliquoted into a new tube and then centrifuged at 10,000 rpm for 5 min to discard the supernatant. One millilitre of 70% methanol was added into the cell pellet and gently resuspended. The cell pellet was collected after centrifugation. All cell types except RBC were washed with 70% methanol. Ten microlitres of 70% methanol were added into the cell pellet and gently mixed until homogeneous. One microlitre of the resuspended cell solution (1 × 10^5^ cells) was dotted on an ITO coated slide and then dried at room temperature. The remaining cells were diluted in two-fold serial dilutions with 70% methanol. Subsequently, 1 µL of varying numbers of cells (100,000, 50,000, 25,000, 12,500, 6250, 3125, 1562 cells, or lower) were dotted on the ITO coated slide, while plasma was diluted in distilled water. Thereafter, 1 µL of the diluted plasma was dotted on an ITO coated slide. After sample dots dried, 1 µL of alpha-cyano-4-hydroxycinnamic acid (HCCA) matrix solution (10 mg/mL HCCA matrix in 50%ACN/0.1%TFA:ACN at a 1:3 ratio) was added to cover a sample dot and dried at room temperature. This step was repeated to acquire a double-layered HCCA matrix covering each sample dot.

### 2.8. MALDI-TOF MS

MALDI-TOF MS was performed using an Ultraflex TOF/TOF mass spectrometer with flexSeries 1.4 and FlexControl version 3.4 software (Bruker Daltonics, Billerica, MA, USA). The nitrogen laser was fired onto the HCCA coated sample spots on ITO coated slides with pulse ion extraction set to 250 ns. Extraction of released ions was accelerated with a voltage of 25 kV in linear positive mode for masses in the range of 5000–20,000 *m/z*. Protein I calibration standard (Bruker Daltonics) was used to calibrate with 500 ppm peak assignment tolerance before sample analysis. Each spectrum is derived from the sum of 5000 laser shots of random-walk partial-sample mode with 2000 Hz frequency and 70% laser power performed in five different regions on a dot of the analysed sample. A signal-to-noise ratio of more than 3 was selected to define peaks with a maximum of 100 peaks per spectrum.

### 2.9. Spectrum Analysis and Database Creation

Each spectrum was analysed in the flexAnalysis version 3.4 software (Bruker Daltonics) to adjust baseline subtraction and smoothing background noise. Signal-to-noise, peak intensity, area under the curve and m/z of each spectrum in each cell type or plasma were exported from flexAnalysis software (Bruker Daltonics) to calculate the average m/z detection in the seven samples of varying cell concentrations. At least 25 spectra per cell type (obtained from five different cell concentrations of each cell type and five spectra per cell concentration) were selected to create specific MSP databases on MALDI Biotyper OC version 3.1 software (Bruker Daltonics). MALDI Biotyper software parameters were optimised and adjusted to create in-house cell-type databases, including frequency threshold for spectra adjusting = 50, frequency threshold for score calculation = 5, a maximum mass error of the raw spectrum = 2000, a mass tolerance of the adjusted spectrum = 500, accepted mass tolerance of a peak = 600, and parameter of the intensity correction function = 0.25. For cell-type identification, the MS spectra derived from 195–100,000 cells of each cell type were predicted. The logarithmic scores were used to evaluate cell-type identification; the higher score means a more confident prediction. The logarithmic scores were presented as mean ± SD. *p*-value < 0.05 is considered statistically significant.

## 3. Results

### 3.1. Profiling of MSP for Various Cell Types Using MALDI-TOF MS

Spectra of normal cell types including RBC, WBC obtained from hMX flow-through, OKT/CD28 T-cell blasts, PBMC, PHA-activated T-cells, and FHs74Int cells, as well as other blood components, i.e., plasma, were detected by MALDI-TOF MS to generate their specific MSP, as shown in Figure 1a. Plasma and RBC spectra showed unique MSP that did not overlap with other cell types. WBC and FHs74Int cells showed similarities in the range of 5000–7000 m/z, but clear differences in the remaining range of spectrum. Similar MSP were observed between PBMC and T-cell blasts after treatment with PHA or OKT3/CD28. However, spectra in the range of 6000–8000 m/z showed either presence/absence or high/low intensities of peaks and therefore still clearly distinguished between those cells (Figure 1a). Next, cancer cell lines, i.e., Jurkat, SK-N-SH, SH-SY5Y, A549, A2780, LNCaP, and Caco-2 cells, were subjected to MALDI-TOF MS and cell-MSP were obtained as shown in Figure 1b. Some MSP at various intensities were common among the different cancer cells, while other peaks were unique. Figure 1c shows the MSP difference between normal and cancer cell lines of the same tissue (intestine—FHs74Int vs. Caco-2 cells; blood—OKT3/CD28 T-cell blast vs. Jurkat cells) and two cancer cell lines of the same tissue but different origin (neuron, SK-N-SH vs. SH-SY5Y). 

### 3.2. Cell-MSP Database Generation

Multiple MSP obtained from normal and cancer cells at a range of cell concentrations (Appendix A) were compiled into the cell-MSP database and analysed by MALDI Biotyper software. Cell-MSP between groups of normal cells versus cancers were compared using MALDI Biotyper software (Bruker), as shown in Figure 2. The corresponding mass peaks were exported to demonstrate the mass-to-charge (m/z), intensity, and frequency, as shown in Appendix A. This finding suggests that cell-MSP acquired by MALDI-TOF MS and classified by MALDI Biotyper software hold promise for detecting atypical cells over a background of normal cells.

### 3.3. Single Cancer Cell Typing

Since the threshold of log score in the MALDI Biotyper software is not available by default for (mammalian) cell-type identification, we optimised parameters and compared cell spectra against the newly generated cell-MSP database for both normal and cancer cells, as described in Material and Methods. All MS spectra generated from 195–100,000 cells of each cell type were predicted by using in-house cell-MSP database. The highest logarithmic score of each cell-MSP correctly predicted the specific cell type, as shown in Table 1. The cell-MSP database was then integrated into the Biotyper software library to further match the spectra of the simulated CAC samples.

### 3.4. Recovery Rates of Cancer Cells after RBC Lysis and WBC Depletion during hMX Separation

We proceeded to validate the performance of the hMX rare-cell isolation platform in our laboratory. Recovery rates of WBC and cancer cell lines after RBC lysis were determined as described in Material and Methods. The average recovery rate for WBC obtained from two to six millilitres of whole blood was found to be 91.2% and ranged from 76.3% to 99.9% for three different cancer cell lines (Figure 3a). WBC depletion was performed on WBC obtained from the RBC lysis procedure (as described in Materials and Methods). hMX WBC depletion reduced ten million WBC to 100,000 to 500,000 in the flow-through of each hMX column, which translates into depletion rates consistently reaching over 95%. At the same time, recovery rates of cancer cell lines passed through hMX columns ranged from 62.7% to 99%, depending on cell line and cell number spiked. Significant differences in recovery rates between cell lines were observed (Figure 3b).

### 3.5. Limit of Detection of Cancer Cells Spiked into WBC

After obtaining the above data, we addressed the question whether MALDI-TOF MS combined with our new in-house cell-MSP library can accurately identify cancer types over a WBC background processed by the hMX system. Blood samples containing rare atypical cells were generated by spiking each of the cancer cell lines (A549, Caco-2, or LNCaP) in decreasing cell numbers (two-fold serial dilutions, i.e., from 100 cells to 3 cells) into 5000 WBC. The simulated CAC samples were then washed with 70% methanol and dotted on ITO slides for MALDI-TOF MS. The cell-MSP of the simulated CAC samples of A549, Caco-2, and LNCaP cell lines were shown in Figure 4.

Their logarithmic scores were exported after matching with the in-house database (Figure 5). The logarithmic scores of the simulated CAC samples (5000 WBC plus cancer cells) as compared to the normal control (5000 WBC) showed a dose-dependent effect with higher number of cancer cells resulting in higher logarithmic scores. MALDI-TOF MS was able to detect A549, Caco-2, and LNCaP cancer cells in the spiked samples down to three, six, and six cells, respectively, over the background noise of 5000 WBC, with significantly higher predictive scores compared to the control (Figure 5).

Finally, we addressed whether this technique could determine the specific cancer types of the simulated CAC samples by the higher log scores. Cell-MSP of simulated CAC samples were then searched against the combined cancer cell-MSP databases, and the log predictive scores were presented in Table 2. As expected, the higher log scores of the simulated CAC samples were matched to their own cancer types with the detection limit down to six cancer cells over the background noise of 5000 WBC.

## 4. Discussion

MALDI-TOF MS already has become an indispensable tool in clinical diagnostics, for example, as a rapid method for bacterial typing in clinical microbiology laboratories [29]. More recently, after generating improved databases, several groups have successfully applied MALDI-TOF MS with MALDI Biotyper software on whole-cell-MSP to profile circulating immune cells [30] and intact mammalian cells [31].

To our knowledge, MALDI-TOF MS has never been optimised to detect tumour-associated circulating rare cells, or in particular, to discriminate circulating cancer cells from circulating blood cells.

From a molecular perspective, MALDI-TOF MS converts cellular peptides, proteins, metabolites, lipids, and nucleic acids into their ionised states, which are recorded as cell-MSP. Although it cannot reveal the identity of any particular molecule due to the limitations of current technology and computational algorithms, MALDI-TOF MS analysis can capture the particular protein profile of each particular cell type. Therefore, these MSP can be used to distinguish different cell types or the same cell type at various states (Figure 1 and Table 1). This allowed us to create the in-house cell-MSP database for MALDI Biotyper software with cluster analysis.

First, we generated an in-house cell-MSP database using MALDI-TOF MS coupled with MALDI Biotyper software, covering plasma, RBC, WBC, activated T-cells and T-cell blasts as well as a range of immortalised cancer cell lines. We found that MSP of all blood components and cell lines were different and that, therefore, we could readily distinguish a variety of blood cells, primary cell lines and cancer cell lines.

Next, we wanted to combine the hMX rare-cell isolation platform previously established in our group with downstream MALDI-TOF MS. We had already demonstrated the platform’s clinical utility for detecting tumour-associated circulating endothelial cells [1]. In this current study, we performed a fresh validation of the platform in our laboratory, which demonstrated consistent depletion of healthy RBC and WBC from heparin blood samples, as well as consistent recovery of cancer cells from cell culture, even down to experiments using only approximately 10 cells. These low-number experiments were performed with A549 cells only because of their robustness and relatively consistent morphology. LNCaP cells, on the other hand, presented with a highly heterogenous morphology and in addition, appear extremely fragile. Caco-2 cells tended to form clusters, even after trypsinization. Lastly, SH-Sy5y cells are blastoma cells and therefore may not reflect the majority of cancers, which are carcinomas.

Nevertheless, while not all cell lines were suitable for low-number experiments in the context of cell separation, we were able to show that the hMX system does not interfere with MALDI-TOF MS in principle: Blood cells processed through the hMX platform do not interfere with the identification of any of the cell lines examined in MALDI-TOF MS, with a limit of detection of only three to six cancer cells in 5000 WBC.

Taken together, our results suggest that whole-cell-MSP deliver highly characteristic signatures which represent particular subtypes of WBC and cancer cells, including their particular physiological or pathological state. This calls for further development of CAC detection using MALDI-TOF MS workflow as a simple, rapid, and economical non-invasive liquid biopsy for cancer screening.

With future advancements in detection sensitivity and resolution of mass spectrometry, we anticipate that a similar strategy as in the present study will be able to routinely detect a single CAC after WBC depletion and potentially even in non-enriched samples, e.g., the buffy coat isolated from 7.5–10 mL whole blood of patients.

However, this study faced several limitations. Although we successfully generated a MALDI-TOF MS database of a wide range of cells and successfully detected spiked cancer cells in WBC, the workflow still lacks true analytical validation which will require spiking of cancer cells into whole blood samples and subsequent detection by MALDI-TOF MS. This next step will be crucial to prepare for the ultimate goal to achieve clinical validation, which is the detection of CAC in blood samples obtained from cancer patients. Nonetheless, working with spiked cancer cell samples in WBC allowed us to understand the limit of detection of cancer cells over a background of WBC (Figure 5), which we consider a first essential step to prepare for the analytical validation process. Importantly, this cannot be performed in patient blood samples because there is no reliable reference test, or gold standard, for detecting CAC. Further challenges include the need to expand the cell-MSP database to cover not only a broader range of cancer cell types but, most importantly, CAC derived from patients with different cancer types and stages.

The latter may eventually enable us to combine multiple MSPs of the same cancer into a diagnostic MSP (that can identify the presence a specific cancer type) and, eventually, multiple MSPs of different cancers into an overarching general cancer screening MSP (that can detect the presence or absence of any CAC in the enriched sample). In both approaches, testing the true clinical accuracy of diagnosing or screening for cancer by MALDI-TOF MS will need to be clinically validated in large, adequately powered and prospectively blinded diagnostic studies.

## 5. Conclusions

In conclusion, this study has established the new CAC detection workflow using MALDI-TOF MS and successfully validated its feasibility by the simulated CAC samples. Further development of the atypical cell-MSP database to cover all cancer cell types sourced from cell lines and patient’s tumours, together with determining the sensitivity and specificity of the workflow in a large patient cohort, will facilitate the use of CAC detection using MALDI-TOF MS as a rapid and economical method for cancer screening in clinical laboratories.

## Figures and Tables

**Figure 1 cancers-13-03775-f001:**
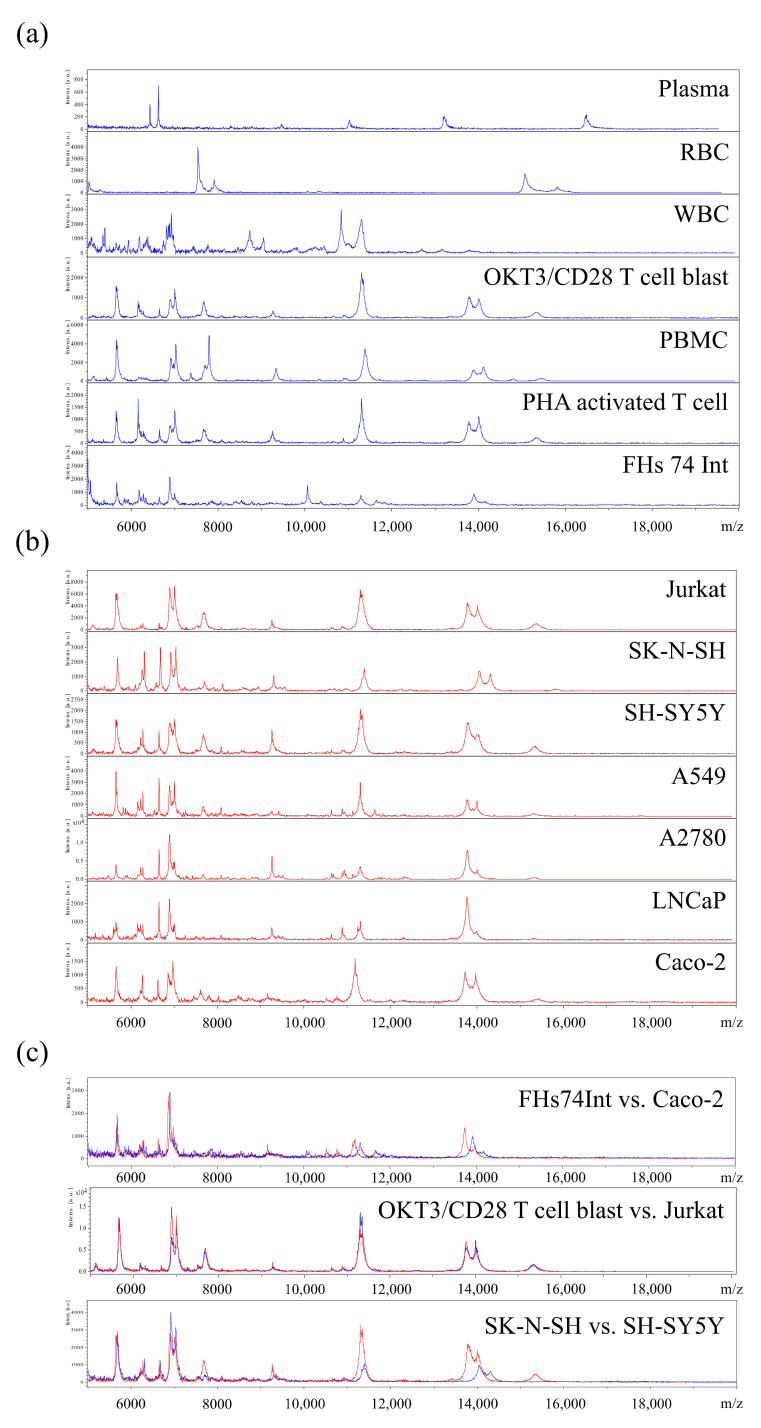
Main spectra profile (MSP) of plasma and various cells. MSP of (**a**) plasma, normal and (**b**) cancer cells. Plasma or individual cell types, respectively, were dotted on the ITO glass slide and processed using MALDI-TOF. (**c**) Overlay of cell-MSP of fetal intestinal cells (FHs74Int) and colon cancer cells (Caco-2), OKT3/CD28 activated T-cell blast and T-cell leukaemia cells (Jurkat), as well as two neuroblastoma cell lines of different origin (SK-N-SH vs. SH-SY5Y) demonstrate that different cell types or the same cell type at different disease or activation states can be distinguished by their cell-MSP pattern.

**Figure 2 cancers-13-03775-f002:**
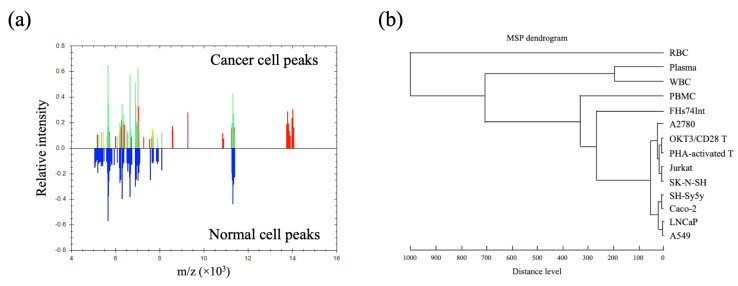
Comparison of normal and cancer cell spectra. (**a**) Integration of individual normal cell spectra and individual cancer cell spectra revealed a set of common and unique peaks with the corresponding mass-to-charge (*m/z*) and relative intensities. Blue peaks are derived from integration of normal cell spectra and red peaks from integration of unique cancer cell spectra. Green and yellow peaks reflect overlapping normal and cancer cell peaks. (**b**) MALDI Biotyper-cluster analysis showing a dendrogram of the MSP of normal and cancer cells. MSP of cancer cells were distinctly separated from normal cells, particularly WBC, the major contaminant in CAC detection.

**Figure 3 cancers-13-03775-f003:**
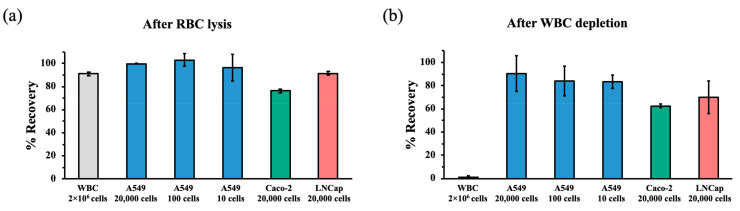
Cell recovery after RBC lysis and WBC depletion. (**a**) For the recovery rate after RBC lysis, two to six millilitres of heparinised whole blood and cancer cell lines resuspended in 1 mL cell culture medium were each subjected to RBC lysis conditions as described in Material and Methods. (**b**) For the recovery rate after WBC depletion, WBC and cancer cells were subjected to hMX WBC cell depletion as described in Material and Methods. Of note, WBC depletion consistently exceeded 95% with an average depletion of 98.7%. All data were obtained from three independent experiments.

**Figure 4 cancers-13-03775-f004:**
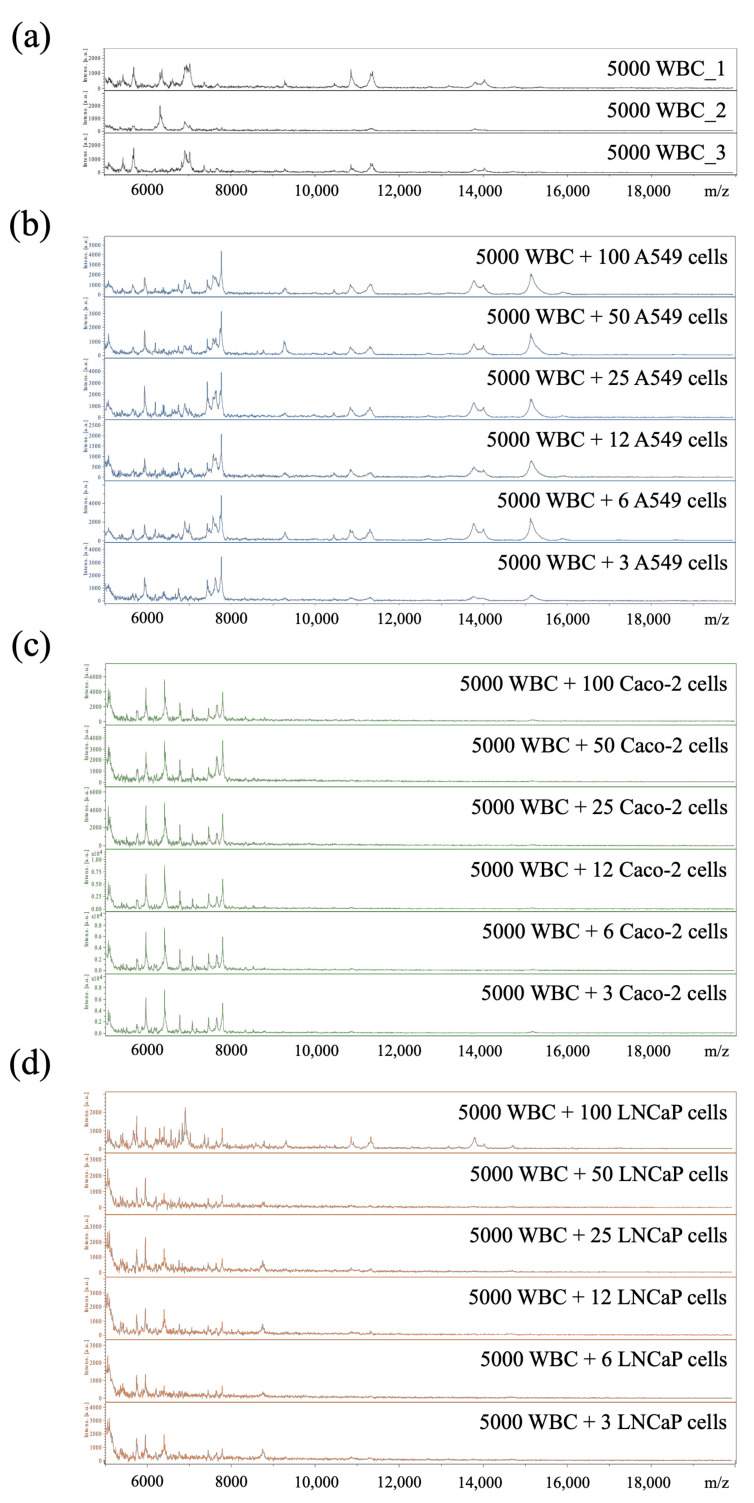
Cell-MSP of simulated CAC samples. Simulated CAC samples were prepared by mixing a constant number of WBC (5000 cells; *n* = 3 individuals) (**a**) and varied cell numbers of cancer cell lines, including A549 (**b**), Caco-2 (**c**), and LNCaP (**d**) (two-fold dilution of 100 to 3 cells) and process using MALDI-TOF MS.

**Figure 5 cancers-13-03775-f005:**
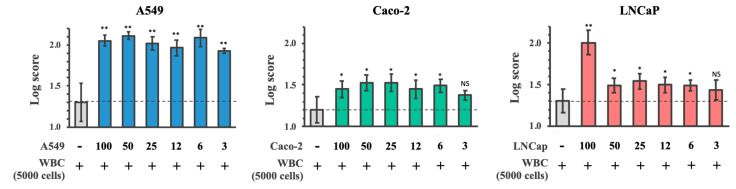
Logarithmic predictive scores of the simulated CAC samples of A549, Caco-2, and LNCaP cell lines. Cell-MSP of the simulated CAC samples, which were prepared by pulsing 3, 6, 12, 25, 50, and 100 cells of A549, Caco-2, or LNCaP into 5000 WBC isolated from three normal individuals, were analysed by MALDI Biotyper software using A549, Caco-2, and LNCaP cell-MSP databases, respectively. The control sample was 5000 WBC without cancer cell pulsing. ** *p* < 0.005 vs. normal; * *p* < 0.05 vs. normal; NS: not significant.

**Table 1 cancers-13-03775-t001:** Logarithmic predictive scores (mean ± SD) of cell-MSP profiles: validation against the newly generated cell-MSP database.

Sample	Normal Condition							Cancers						
	FHs74Int	PBMC	PHA-Activated T-Cell	Plasma	RBC	OKT/CD28 T-Cell Blast	WBC	A2780	A549	Caco-2	Jurkat	LNCaP	SK-N-SH	SH-SY5Y
FHs74Int	**2.1350 ± 0.2803**	1.4915 ± 0.1994	1.6322 ± 0.3051	1.6348 ± 0.1322	0.6119 ± 0.3286	1.7119 ± 0.2587	1.8191 ± 0.0550	1.9049 ± 0.1933	2.0821 ± 0.1652	2.0012 ± 0.1727	1.7303 ± 0.169	2.0619 ± 0.1925	1.7640 ± 0.2299	1.8670 ± 0.2323
PBMC	1.7701 ± 0.1779	**2.1496 ± 0.2651**	1.4963 ± 0.2915	1.3458 ± 0.3576	0.5609 ± 0.4154	1.5627 ± 0.3147	1.6248 ± 0.1681	1.6007 ± 0.3079	1.5321 ± 0.3421	1.6986 ± 0.3045	1.7353 ± 0.3831	1.5169 ± 0.2545	1.5917 ± 0.3260	1.7939 ± 0.2621
PHA-activated T-cell	2.0488 ± 0.1909	1.6674 ± 0.1591	**2.6313 ± 0.1089**	0.7704 ± 0.1640	0.5730 ± 0.3678	2.5584 ± 0.1147	1.5338 ± 0.1197	2.1848 ± 0.1307	2.1576 ± 0.1746	2.1501 ± 0.1913	2.3600 ± 0.1887	2.1788 ± 0.0780	2.3289 ± 0.1192	2.3708 ± 0.0889
Plasma	1.3979 ± 0.2461	0.9940 ± 0.1887	0.7743 ± 0.2208	**1.8753 ± 0.1970**	0.4516 ± 0.3364	0.6832 ± 0.2934	1.7469 ± 0.1210	1.0028 ± 0.2665	1.1963 ± 0.3093	1.1208 ± 0.2618	0.9300 ± 0.2648	1.1220 ± 0.1533	0.8733 ± 0.2831	1.0638 ± 0.1367
RBC	0.5175 ± 0.6793	0.7217 ± 0.4881	0.2508 ± 0.2748	0.5910 ± 0.6671	**1.9847 ± 0.3313**	0.1100 ± 0.2578	0.7912 ± 0.6389	0.1550 ± 0.3025	0.5266 ± 0.5596	0.4926 ± 0.4792	0.3714 ± 0.2815	0.5056 ± 0.6176	0.2286 ± 0.3102	0.4845 ± 0.4842
OKT/CD28 T-cell blast	2.0774 ± 0.1380	1.6846 ± 0.0643	2.5799 ± 0.0458	0.8026 ± 0.2365	0.7324 ± 0.3286	**2.7106 ± 0.0734**	1.5184 ± 0.0869	2.2299 ± 0.0837	2.2066 ± 0.1191	2.1997 ± 0.1279	2.4063 ± 0.1366	2.2629 ± 0.0394	2.2910 ± 0.1400	2.4191 ± 0.0642
WBC	1.4917 ± 0.2092	1.1169 ± 0.3018	0.9202 ± 0.4080	1.6827 ± 0.2131	0.5963 ± 0.1684	0.8511 ± 0.2450	**2.3397 ± 0.1838**	1.1211 ± 0.2700	1.3692 ± 0.2490	1.2038 ± 0.1767	1.1408 ± 0.2247	1.3104 ± 0.2443	1.0331 ± 0.2621	1.1588 ± 0.2317
A2780	2.1235 ± 0.1602	1.5215 ± 0.1751	2.0660 ± 0.2124	1.2680 ± 0.3177	0.3727 ± 0.3457	2.1136 ± 0.2224	1.6691 ± 0.0992	**2.4944 ± 0.3423**	2.2643 ± 0.1459	2.2756 ± 0.2128	2.2494 ± 0.2313	2.2714 ± 0.1548	2.2019 ± 0.2970	2.2298 ± 0.2702
A549	2.1816 ± 0.2151	1.5275 ± 0.1524	1.8982 ± 0.2033	1.5506 ± 0.2227	0.3753 ± 0.3012	1.9355 ± 0.1950	1.6936 ± 0.1122	2.1782 ± 0.2156	**2.4265 ± 0.1411**	2.1329 ± 0.1696	2.0331 ± 0.2179	2.2404 ± 0.1166	2.0062 ± 0.1448	2.1106 ± 0.1536
Caco-2	1.9453 ± 0.1441	1.5742 ± 0.2473	1.9038 ± 0.2818	1.3304 ± 0.2509	0.4539 ± 0.2892	1.9255 ± 0.3473	1.8031 ± 0.0916	1.9760 ± 0.2606	2.0722 ± 0.1635	**2.3448 ± 0.1935**	1.9798 ± 0.2957	2.0498 ± 0.1952	1.9230 ± 0.2567	1.9398 ± 0.3225
Jurkat	1.9232 ± 0.1248	1.7166 ± 0.1735	2.1508 ± 0.1942	1.0892 ± 0.3198	0.6737 ± 0.3454	2.1828 ± 0.1656	1.5729 ± 0.0996	2.1918 ± 0.1368	2.1099 ± 0.0933	2.2634 ± 0.1560	**2.4694 ± 0.2012**	2.1695 ± 0.1454	2.2253 ± 0.2057	2.2867 ± 0.1489
LNCaP	2.0905 ± 0.1492	1.4469 ± 0.1570	2.0563 ± 0.1790	1.3670 ± 0.3046	0.6243 ± 0.2958	2.1313 ± 0.1589	1.7233 ± 0.1455	2.2517 ± 0.1628	2.2276 ± 0.1262	2.1828 ± 0.1285	2.2080 ± 0.2369	**2.5943 ± 0.1017**	2.1270 ± 0.1728	2.1682 ± 0.1521
SK-N-SH	1.9944 ± 0.3021	1.7178 ± 0.2224	2.0928 ± 0.3945	1.1005 ± 0.2506	0.6191 ± 0.5308	2.1130 ± 0.4056	1.5143 ± 0.1422	2.1351 ± 0.3092	1.9520 ± 0.4005	2.1358 ± 0.2139	2.2001 ± 0.3488	1.9942 ± 0.4116	**2.2406 ± 0.4680**	2.2211 ± 0.2660
SH-SY5Y	2.0903 ± 0.1268	1.7604 ± 0.1654	2.2288 ± 0.1694	1.1884 ± 0.3393	0.7160 ± 0.5143	2.2949 ± 0.2257	1.6132 ± 0.1342	2.2423 ± 0.1941	2.1589 ± 0.1547	2.2678 ± 0.1251	2.3613 ± 0.2017	2.2732 ± 0.0793	2.2930 ± 0.1805	**2.4820 ± 0.1413**

Bold with the background red colour represents the highest log score of validation against cell-MSP database.

**Table 2 cancers-13-03775-t002:** Logarithmic predictive scores (mean ± SD) of WBC spiked with various cell numbers of A549, Caco-2, and LNCaP cancer: Validation against seven cancer cell-MSP databases.

Sample	Cancers						
	A2780	A549	Caco-2	Jurkat	LNCaP	SK-N-SH	SH-SY5Y
WBC alone (5000 cells)	1.0321 ± 0.2761	1.2988 ± 0.2297	1.2059 ± 0.1572	0.9033 ± 0.1858	1.3054 ± 0.1392	0.8873 ± 0.2569	1.1791 ± 0.1741
WBC + A549 100 cells	1.9630 ± 0.1058	**2.0542 ±** **0.0675**	1.9288 ± 0.0837	1.9156 ± 0.0648	2.0164 ± 0.1132	1.9640 ± 0.1252	1.9562 ± 0.1576
WBC + A549 50 cells	2.0150 ± 0.0131	**2.1133 ±** **0.0446**	2.0340 ± 0.0342	1.8923 ± 0.0985	2.0093 ± 0.0276	2.0413 ± 0.0614	2.0740 ± 0.0288
WBC + A549 25 cells	1.8237 ± 0.1525	**2.0170 ±** **0.0832**	1.8877 ± 0.1299	1.8937 ± 0.0535	1.9943 ± 0.0488	1.9313 ± 0.0803	1.9473 ± 0.1320
WBC + A549 12 cells	1.7277 ± 0.1466	**1.9660 ±** **0.0957**	1.7647 ± 0.1791	1.6070 ± 0.1327	1.8267 ± 0.1426	1.5607 ± 0.0715	1.7387 ± 0.1296
WBC + A549 6 cells	2.0620 ± 0.1580	**2.0867 ±** **0.1100**	1.9493 ± 0.1955	1.9973 ± 0.0304	2.0787 ± 0.1186	2.0723 ± 0.0509	2.1767 ± 0.0850
WBC + A549 3 cells	1.6043 ± 0.1917	**1.9287 ±** **0.0325**	1.6313 ± 0.1517	1.4933 ± 0.1420	1.6990 ± 0.1413	1.5620 ± 0.1274	1.6447 ± 0.0674
WBC + Caco-2 100 cells	0.9962 ± 0.2365	1.2082 ± 0.1990	**1.4487 ±** **0.0973**	0.9223 ± 0.1924	1.1708 ± 0.0903	0.9162 ± 0.2979	1.1570 ± 0.1816
WBC + Caco-2 50 cells	1.1342 ± 0.1317	1.2037 ± 0.0660	**1.5260 ±** **0.0979**	1.0262 ± 0.1944	1.3920 ± 0.1757	0.9752 ± 0.1857	1.2365 ± 0.0965
WBC + Caco-2 25 cells	1.1372 ± 0.2578	1.3704 ± 0.1549	**1.5273 ±** **0.1048**	1.1706 ± 0.2425	1.3114 ± 0.1364	1.0816 ± 0.2680	1.1946 ± 0.1729
WBC + Caco-2 12 cells	1.0020 ± 0.2398	1.1852 ± 0.3041	**1.4497 ±** **0.1100**	1.0293 ± 0.1586	1.2397 ± 0.1651	0.8930 ± 0.1938	1.1543 ± 0.1732
WBC + Caco-2 6 cells	1.2602 ± 0.1989	1.4042 ± 0.1298	**1.4900 ±** **0.0757**	1.2568 ± 0.2329	1.3110 ± 0.1401	0.9445 ± 0.1940	1.3058 ± 0.2022
WBC + Caco-2 3 cells	1.1419 ± 0.2372	1.2013 ± 0.1595	**1.3747 ±** **0.0539**	1.0453 ± 0.3087	1.1704 ± 0.1621	0.8650 ± 0.2197	1.2226 ± 0.1806
WBC + LNCaP 100 cells	1.9282 ± 0.1325	1.7682 ± 0.1501	1.7817 ± 0.2241	1.7393 ± 0.2260	**2.0077 ±** **0.1448**	1.7920 ± 0.2132	1.7323 ± 0.2172
WBC + LNCaP 50 cells	1.2056 ± 0.1210	1.4430 ± 0.0933	1.2372 ± 0.1781	0.9994 ± 0.0773	**1.4907 ±** **0.0906**	0.9514 ± 0.2849	1.2144 ± 0.1610
WBC + LNCaP 25 cells	1.1788 ± 0.1377	1.4313 ± 0.0577	1.2675 ± 0.2249	1.0938 ± 0.1024	**1.5390 ±** **0.0897**	0.8865 ± 0.1607	1.3278 ± 0.0426
WBC + LNCaP 12 cells	1.0170 ± 0.2486	1.2545 ± 0.1651	1.1175 ± 0.1503	1.0415 ± 0.1445	**1.4963 ±** **0.0976**	0.9748 ± 0.1699	1.1278 ± 0.2319
WBC + LNCaP 6 cells	1.1422 ± 0.1773	1.2156 ± 0.2179	1.1712 ± 0.0925	0.9804 ± 0.1444	**1.4910 ±** **0.0646**	1.0528 ± 0.1320	1.2680 ± 0.1429
WBC + LNCaP 3 cells	1.0447 ± 0.2740	**1.4350 ±** **0.1640**	1.1595 ± 0.1455	0.9908 ± 0.1840	1.4340 ± 0.1204	0.8315 ± 0.1774	1.1303 ± 0.2518

Bold with the background red colour represents the highest log score of validation against cell-MSP database.

## Data Availability

The data presented in this study are contained within the article and the Appendix A. Raw cell-MSP are available from the corresponding author (S.C.) upon request.

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
