# Peer review of "Cell-Main Spectra Profile Screening Technique in Simulation of Circulating Tumour Cells Using MALDI-TOF Mass Spectrometry"

_cancers, 2021, doi:10.3390/cancers13153775_

Round 1
Reviewer 1 Report
The manuscript from Chiangjong et al. presents a MALDI-TOF MS platform for detecting the circulating tumor cells (CTC) under the background of white blood cells (WBC). They constructed the cell-main spectra profile (MSP) for each of the selected cancer cells and normal blood cells and evaluated the limit of detection in this approach. This work is interesting and could be important for the clinical cancer diagnosis if sufficient sensitivity is reached. However, several issues should be further revised or addressed before publication. Following are comments and suggestions:
- In general, the resolution of the figures, especially the MS spectra, are quite poor. Please label the X-axis more clearly. It would be better to re-organize the figures for better reading.
- Figure 1: The authors mentioned that they created the cell-specific MSP database by selecting 5 MS spectra that came from 5 different concentrations per cell type. Please provide more details on how to select and integrate these 5 MS spectra to create the MSP since we can observe very diverse profile in observed peaks and their relative intensities in each cell type with different concentrations (in Figure S1). If I compare the MSP in Figure 1 and Fig S1B for WBC, it looks like the MSP of WBC in Figure 1 is the same as the MS spectra from 6250 WBCs.
- Figure 2: There is a clear separation of MSP between normal and cancer cells. Figure 2b suggests that the MSP between A549 and LNCaP as well as Caco-2 and SH-Sy5y are more similar. Is there any patten found in cells with similar morphology??
- Table 1: please clarify how to determine the threshold of log score in the MALDI Biotyper software by optimizing parameters. It’s not clear that how many cells are analyzed by MALDI-TOF in each cell type to get the score in Table 1 since different cell number would generate different MS profile (Fig S1).
- Figure 3: the author evaluated the recovery of cells after RBC Lysis and WBC Depletion during Hmx Separation. They started the experiments separately with 9000-20000 cancer cells for RBC lysis and 5000-10000 cancer cells for WBC depletion which is a high cell number compared to that in patient’s blood. Lower number of starting cells may cause significant loss during the experiment. Thus, it is critical to check whether lower number of cancer cells can be recovered with same efficiency. In addition, the author should have an experiment to mix RBC and WBC with a defined number of cancer cells (just mimic the true patient’s blood) and then perform Hmx Separation to evaluate the real recovery of cancer cells.
- Figure 4: (a) the MS spectra of 5000 WBCs from 3 individuals are quite different. Would this background difference affect the identification of the cancer cells by using MSP?? It’s also not clear how the authors prepared the 5000 WBCs for mixing with different number of cancer cells. A mixture of the WBCs from 3 individuals or the WBCs isolated from one of the individuals was used??
- Figure 5 and Table 2: There is a significant difference between WBC alone and the simulated CAC samples (Figure 5). The authors claimed that the higher log scores of the simulated CAC samples were matched to their own cancer types (Table 2), but whether the obtained score in each cancer type would have significant difference when comparing to the other cancer cells??
- Although the authors mentioned that they did not use the blood samples from cancer patients in current stage as the limitations. I think it’s worth to try to spike different number of cancer cells in the blood sample and run the whole experiment to evaluate the analytical performance of current method.
Other suggestions:
- Please prepare a content information in the first page of supplementary materials.
- “foetal” bovine serum should be “fetal”.
- Line 183: phosphate-buffered saline containing 5 mM ethylenediaminetetraacetic acid and 1% foetal bovine serum (PBS/EDTA 5 mM/FBS 1%). I think it’s okay to delete the full names and directly use abbreviations (PBS/EDTA 5 mM/FBS 1%) for these chemicals because they have been described in the earlier paragraph.
- Line 267: WBC and “FHs 74 Int” cells. Please revise “FHs 74 Int” to “FHs74Int” in order to keep consistency for this term.
Author Response
Reviewer#1
The manuscript from Chiangjong et al. presents a MALDI-TOF MS platform for detecting the circulating tumor cells (CTC) under the background of white blood cells (WBC). They constructed the cell-main spectra profile (MSP) for each of the selected cancer cells and normal blood cells and evaluated the limit of detection in this approach. This work is interesting and could be important for the clinical cancer diagnosis if sufficient sensitivity is reached. However, several issues should be further revised or addressed before publication. Following are comments and suggestions:
We are grateful that the reviewer found the cell-MSP technique is interesting and could be important for the field of cancer diagnosis. We would like to thank this reviewer for positive comments and helpful suggestions. Please find below the point-by-point response to the reviewer’s comments, in which all corresponding changes in the manuscript are highlighted in red color texts.
1. In general, the resolution of the figures, especially the MS spectra, are quite poor. Please label the X-axis more clearly. It would be better to re-organize the figures for better reading.
Response: Thank you for this helpful suggestion. The resolution of the figures at X-axis was improved. Also, the figures were re-organized for better reading.
2. Figure 1: The authors mentioned that they created the cell-specific MSP database by selecting 5 MS spectra that came from 5 different concentrations per cell type. Please provide more details on how to select and integrate these 5 MS spectra to create the MSP since we can observe very diverse profile in observed peaks and their relative intensities in each cell type with different concentrations (in Figure S1). If I compare the MSP in Figure 1 and Fig S1B for WBC, it looks like the MSP of WBC in Figure 1 is the same as the MS spectra from 6250 WBCs.
Response: The cell-specific MSP database of each cell type was created from all MS spectra of all respective concentrations. One dot of one cell concentration on the ITO coated slide generated 5 MS spectra. Twenty-five MS spectra from five different cell concentrations of each cell type were integrated (S/N>3 and intensity>500) into one cell-specific MSP database. The MS spectra of each cell type in Figure 1 is representative of the MS spectra in Figure S1.
3. Figure 2: There is a clear separation of MSP between normal and cancer cells. Figure 2b suggests that the MSP between A549 and LNCaP as well as Caco-2 and SH-Sy5y are more similar. Is there any pattern found in cells with similar morphology??
Response: The authors appreciate the reviewer’s question, this is quite intriguing indeed. The morphology of all four cell lines (A549, LNCaP, Caco-2, SH-SY5Y) appears very different indeed: A549 cells (lung cancer) are relatively homogenous for a cancer cell line. They appear fast growing, robust, with high N/C ratio and overall a low degree of multinucleation. LNCaP (prostate cancer), on the other hand, are slow growing, appear extremely inhomogenous, with very large, partly multinucleated cells all the way down to much smaller cells. In addition LNCaP are extremely fragile in our hands, with a considerable number of cells broken, lots of cell debris and naked nuclei scattered over slides (after cytocentrifugation). Caco-2 cells (colon cancer) are less homogenous than A549 but more so than LNCaP, and also much more robust than the latter. Lastly, SH-SY5Y, as a blastoma cell line, much unlike the other three cell lines, are cytokeratin-negative but brightly positive for vimentin (mesenchymal antigen), fast growing and relatively homogenous. To sum, a lot more research is necessary to understand how these properties translate into MSP, to discover patterns and finally, to concisely answer the reviewer’s question! For the purpose of the present study, we can only say with confidence that WBC and cancer cell line MSP are clearly different, even at very low cancer cell numbers over a background of WBC.
4. Table 1: please clarify how to determine the threshold of log score in the MALDI Biotyper software by optimizing parameters. It’s not clear that how many cells are analyzed by MALDI-TOF in each cell type to get the score in Table 1 since different cell number would generate different MS profile (Fig S1).
Response: The authors’ apologize for any unclarity. MALDI Biotyper software contained bacteria database and provides default parameter for bacteria identification. In a first trial, the cell-specific MSP database was generated using default parameters. It identified the MS spectra of each cell type, but it returned low log-scores. Therefore, we optimized the parameters for cell-type identification as explained in the Materials and Methods section. The log-scores in Table 1 were generated from the log-scores of all MS spectra (195-100,000 cells) of each cell type. This clarification was added to the Materials and Methods (page 6, line 263-264) and the Results (page 8, line 314-315).
5. Figure 3: the author evaluated the recovery of cells after RBC Lysis and WBC Depletion during Hmx Separation. They started the experiments separately with 9000-20000 cancer cells for RBC lysis and 5000-10000 cancer cells for WBC depletion which is a high cell number compared to that in patient’s blood. Lower number of starting cells may cause significant loss during the experiment. Thus, it is critical to check whether lower number of cancer cells can be recovered with same efficiency.
Response: The authors agree that this is of concern. We performed an additional experiment examining recovery rates of 100 and 10 A549 cancer cells after each module, respectively. The data is added in results (Figure 3). Material and Methods (page 5, line 210-214) has been updated accordingly. We also added sentences in the Discussion (page 14, line 405-411) to explain our choice of A549 cell line for this particular experiment.
In addition, the author should have an experiment to mix RBC and WBC with a defined number of cancer cells (just mimic the true patient’s blood) and then perform Hmx Separation to evaluate the real recovery of cancer cells.
The authors agree that this will be an essential experiment, however, please note that the present study is a first pilot of combining negative isolation of atypical cells with MALDI-TOF, with the objective to demonstrate that MALDI-TOF can be used to detect low numbers of cancer cells over a background of WBC processed by hMX technology. Towards the end of the discussion, we had already mentioned that “….the workflow lacks true clinical validation….” (in the previous version of the manuscript). Taking into account the reviewer’s comment, we expanded this to “ … the workflow still lacks true analytical validation which will require spiking of cancer cells into whole blood samples and subsequent detection by MALDI-TOF MS. This next step will be crucial to prepare for the ultimate goal to achieve clinical validation, which is the detection of CAC in blood samples obtained from cancer patients.” (page 14, line 428-431). We believe this now adequately reflects the fact that the current paper presents a pilot study, and that subsequent (ongoing) work is required for complete diagnostic method validation.
6. Figure 4: (a) the MS spectra of 5000 WBCs from 3 individuals are quite different. Would this background difference affect the identification of the cancer cells by using MSP?? It’s also not clear how the authors prepared the 5000 WBCs for mixing with different number of cancer cells. A mixture of the WBCs from 3 individuals or the WBCs isolated from one of the individuals was used??
Response: This is an important comment. We used WBC isolated from one individual. The difference of WBC populations between individuals WBCs did not affect the identification of the cancer types in our experience. We did not pool WBC because i) there are different MSP patterns of WBCs among individuals which may depend on various proportions of neutrophils, eosinophils, basophils, monocytes and lymphocyte subsets; ii) in a true diagnostic situation, the CACs will be isolated from an individual.
Note that we added an additional paragraph to Materials and Methods (page 5, line 215-223) to address this as well: “2.6. Spiked Cancer Cell/WBC Sample Preparation. WBC from an individual obtained from hMX flow-through were counted and diluted in 70% methanol with a final concentration of 5,000 cells/μl. Cancer cells including A549, Caco-2, and LNCaP, were counted and diluted in 70% methanol in serial dilution with final concentrations of 100, 50, 25, 12, 6, and 3 cells/μl. These serially diluted cancer cells were spiked into WBCs (5,000 cells/μl) before dotting on the ITO coated slide. Serial dilutions were performed from stock solution. Aliquots from each serial dilution were counted in triplicate. A forth aliquot of the respective serial dilution was used in the experiment.”
7. Figure 5 and Table 2: There is a significant difference between WBC alone and the simulated CAC samples (Figure 5). The authors claimed that the higher log scores of the simulated CAC samples were matched to their own cancer types (Table 2), but whether the obtained score in each cancer type would have significant difference when comparing to the other cancer cells??
Response: Thank you for your comments. Due to the fact that MS spectra pattern of each cancer cell types were close to each other as shown in MSP dendrogram in Figure 2b, the logarithmic predictive scores of simulated CAC samples was able to clearly distinguish cancer cells from WBC but still similar to other cancer cell types. Nonetheless, we observed that the highest predictive scores matched to each respective cancer type at the CAC detectable levels (6-100 cancer cells). This will serve as a foundation for improvement in future studies.
8. Although the authors mentioned that they did not use the blood samples from cancer patients in current stage as the limitations. I think it’s worth to try to spike different number of cancer cells in the blood sample and run the whole experiment to evaluate the analytical performance of current method.
Response: This is duly noted, however, please also see our response to point 5 (second part). We would like to re-iterate that the present study is a pilot study to demonstrate the feasibility of using MALDI-TOF technology to detect cancer cells in blood samples processed by hMX technology. It is not set up as an analytical or even clinical validation of a novel diagnostic method. While we fully agree that these are crucial next steps, both analytical and clinical validations are considerably large projects. We believe, when performed according to international guidelines, these will certainly warrant separate publications. As mentioned in our response to point 5, we have expanded the limitations section of the discussion to elaborate on this fact in greater detail.
Other suggestions:
Thank you for your suggestion. We corrected them as you can see in text.
1. Please prepare a content information in the first page of supplementary materials.
Response: The content information was added to the first page of supplementary materials.
2. “foetal” bovine serum should be “fetal”.
Response: “foetal” was corrected to “fetal”.
3. Line 183: phosphate-buffered saline containing 5 mM ethylenediaminetetraacetic acid and 1% foetal bovine serum (PBS/EDTA 5 mM/FBS 1%). I think it’s okay to delete the full names and directly use abbreviations (PBS/EDTA 5 mM/FBS 1%) for these chemicals because they have been described in the earlier paragraph.
Response: We revised it as the reviewer’s suggestion.
4. Line 267: WBC and “FHs 74 Int” cells. Please revise “FHs 74 Int” to “FHs74Int” in order to keep consistency for this term.
Response: We corrected it.
Reviewer 2 Report
In the manuscript Chiangjong et al. adress a lack of early cancer diagnosis by the detection of circulating tumor cells via MALDI-TOF MS. With this topic the authors work on a current and important aspect. Using highly sensitive mass spec methods for the detection of few cells from blood the great potential of minimal invasive and fast cancer detection methods can be reached in future. This work is one good step in this direction.
The manuscript is well structured and written which makes it comfortable in understanding for a wide audience. Also the experiments and the presented results are respectable. I strongly recommend publication of this work. Indeed some minor revisions should be done.
Introduction:
In the introduction a part fort he classification or the different subtypes of circulating cells relevant for cancer diagnosis should be implemented. It is not really clear, whether the authors discriminate between circulating tumour cells and circulation atypical cells. Or is this the same?
In l. 170 the „A“ in MALDI is „assisted“ not „associated“.
Methods:
For sample preperation serial diluations were used. With this method the resultant cell number can only be estimated. Therefore numbers and the joined statement that 6 cells could be detected can not be made. The authors should redraft these statements and comment on this. For future experiments another sample preperation method should be used and cell counting should not be done with a haemocytometer.
The samples were dotted on ITO coated glass slides. With the Bruker instruments you have the possiblity to use steel target plates. This allows are standardized sample preperation and therefore reproducible measurements. In future experiments this should be done.
In general the authors should clearly state, that they are using the Biotyper software and not the instrument itself. Sometimes this is confusing, for example: ll. 289-290 here it said, that they analyzed with Biotyper, but only the software was used, this should be more clear.
Author Response
Reviewer#2
In the manuscript Chiangjong et al. address a lack of early cancer diagnosis by the detection of circulating tumor cells via MALDI-TOF MS. With this topic the authors work on a current and important aspect. Using highly sensitive mass spec methods for the detection of few cells from blood the great potential of minimal invasive and fast cancer detection methods can be reached in future. This work is one good step in this direction.
The manuscript is well structured and written which makes it comfortable in understanding for a wide audience. Also the experiments and the presented results are respectable. I strongly recommend publication of this work. Indeed some minor revisions should be done.
We are pleased that the reviewer found this work has the potential to be one of the minimal invasive and fast cancer detection methods in future. We thank the reviewer for positive comments and helpful suggestions. Please find below the point-by-point response to all reviewer’s comments.
Introduction:
In the introduction a part for the classification or the different subtypes of circulating cells relevant for cancer diagnosis should be implemented. It is not really clear, whether the authors discriminate between circulating tumour cells and circulation atypical cells. Or is this the same?
Response: We thank the reviewer for this point. We are using the term CAC as an umbrella term for all atypical cancer-associated circulating cells in the blood circulation (including classical epithelial CTC, tCEC or any other subtype of cancer-associated cell). We added a sentence to clarify this point in the Introduction (page 2, line 61-63).
In l. 170 the “A“ in MALDI is “assisted“ not “associated“.
Response: We corrected it.
Methods:
For sample preperation serial diluations were used. With this method the resultant cell number can only be estimated. Therefore numbers and the joined statement that 6 cells could be detected can not be made. The authors should redraft these statements and comment on this. For future experiments another sample preperation method should be used and cell counting should not be done with a haemocytometer.
Response: Thank you for your comments. The cell counting using haemacytometer was used as a method to count cell concentration of the stock cell suspension. Serial dilutions were then performed from this stock solution. Aliquots from each serial dilution were counted in triplicate. With sufficient practice, we found it quite possible to achieve reasonable consistency of these triplicate counts, even for very low cell counts. Once this consistency was achieved, a forth aliquot of the respective serial dilution was used in the experiment. Consistency of this data is also confirmed by the fact that the amount of protein reflected by MS spectra protein pattern correlates well with cell numbers. We added a paragraph in Material and Methods (page 5, line 215-223) to explain our approach of triplicate counting and spiking the fourth aliquot. Notwithstanding this response, we note the reviewer’s concern and going forward, counting of very low cell counts will be performed in the wells of microtiter plates on an inverted fluorescence microscope which has recently become available to our group.
The samples were dotted on ITO coated glass slides. With the Bruker instruments you have the possibility to use steel target plates. This allows are standardized sample preperation and therefore reproducible measurements. In future experiments this should be done.
Response: Thank you for your suggestion. The property of ITO coated glass slide and the steel target plate is similar. However, ITO coated glass slide was selected to use in this study because it is easier to handle and inexpensive compared to the steel target plate. Therefore, it should contribute to making our protocol reproducible between labs, and also enable peripheral labs to send samples to central labs with MALDI-TOF equipment. Also, the ITO coated glass slides are commercially available which facilitates access.
In general the authors should clearly state, that they are using the Biotyper software and not the instrument itself. Sometimes this is confusing, for example: ll. 289-290 here it said, that they analyzed with Biotyper, but only the software was used, this should be more clear.
Response: Thank you for your suggestion. The Biotyper was corrected to the Biotyper software throughout the manuscript.
Round 2
Reviewer 1 Report
The authors have adequately responded to all my questions.